# A phase I study of intra-anal artesunate (suppositories) to treat anal high-grade squamous intraepithelial lesions

Sandy Hwang Fang[1]*, Mihaela Plesa[2], Evie H. Carchman[3], Nicole A. Cowell[4], Emily Staudt[3], Kyleigh Ann Twaroski[3], Ulrike K. Buchwald[5], Cornelia L. Trimble[6]

1 Department of Surgery, Oregon Health & Science University, Portland, Oregon, United States of America, 2 Frantz Medical Development, Ltd, Mentor, Ohio, United States of America, 3 Department of Surgery, University of Wisconsin School of Medicine, Madison, Wisconsin, United States of America, 4 Department of Surgery, Johns Hopkins University School of Medicine, Baltimore, Maryland, United States of America, 5 Department of Medicine, Division of Gastroenterology and Hepatology, Johns Hopkins University School of Medicine, Baltimore, Maryland, United States of America, 6 Department of Obstetrics and Gynecology, Johns Hopkins University School of Medicine, Baltimore, Maryland, United States of America

* fangs@ohsu.edu

**Data Availability Statement:** All relevant data are within the paper and its Supporting Information files.

## Abstract

### Background

Ablation or surgical excision is the typical treatment of anal high-grade squamous intraepithelial lesions (HSIL). Recurrences are common due to the persistence of underlying human papillomavirus (HPV) infection. Additional well-tolerated and effective non-surgical options for HPV-associated anal disease are needed.

### Methods

This 3+3 dose escalation Phase I clinical trial evaluated the safety and tolerability of artesunate suppositories in the treatment of patients with biopsy-proven HSIL.

### Results

The maximal tolerated dose was 400 mg, administered in 3 cycles. All adverse events associated with the use 200- and 400-mg artesunate suppositories were Grade 1. At the 600-mg dose, patients experienced clinically significant nausea.

### Conclusion

Artesunate suppositories are a safe treatment option for anal HSIL.

## Background

Up to 95% of anal cancers (anal squamous cell cancers, ASCC) are caused by persistent infection with human papillomaviruses (HPVs) [1]. The randomized Anal Cancer/HSIL Outcomes

**Funding:** The first author and principal investigator received funding for salary and research support, from Frantz Viral Therapeutics. Frantz Viral Therapeutics, LLC designed, formulated, and supplied the artesunate suppositories, which they provided free of charge, and provided partial financial support for study management and specimen processing. Dr. Trimble's institution reports grants from Frantz Viral Therapeutics, LLC (FVT), for the conduct of the study. She is on the Scientific Advisory Board for FVT. Ms. Plesa reports personal fees from Frantz Viral Therapeutics, LLC during the conduct of the study. She works full-time as Director of Research Programs for FVT. Ulrike K Buchwald is an employee of Merck Sharp & Dohme LLC., a subsidiary of Merck & Co., Inc., Rahway, NJ, USA and may own stock and/or stock options in Merck & Co., Inc., Rahway, NJ, USA. Please note that while Dr. Buchwald is employed and has stock and/or stock options with Merck, Merck did not fund any portion of the study described in this manuscript. The specific roles of these authors are articulated in the 'author contributions' section.

**Competing interests:** The competing interest exists in the fact that Mihaela Plesa became the Director of Research at Frantz Viral Therapeutics when she left Johns Hopkins Hospital. Please note that we had already written the protocol, obtained FDA and IRB approval, and started the study prior to Ms. Plesa leaving Hopkins to join Frantz Viral Therapeutics. This does not alter our adherence to PLOS ONE policies on sharing data and materials.

Research (ANCHOR) study, recently reported that treatment of HPV-related precancerous high-grade anal dysplasia (anal high-grade squamous intraepithelial lesion, HSIL) significantly reduced the incidence of anal cancer as compared to active monitoring [2]. Management of HSIL typically consists of ablation with electrocautery or infrared coagulation in the office or operating room, surgical excision, and topical treatment with immunomodulatory or cytotoxic agents such as imiquimod and 5-fluorouracil (5FU), respectively. Local recurrence or new, metachronous anal HSIL may occur with all treatment modalities, especially as the underlying persistent HPV infection may not be cured. Smoking, poor HIV control (low CD4 cell count or HIV viremia), persistent infection with HPV16/18 or chronic mixed HPV infections can increase the risk for recurrent disease [2–4]. Novel approaches to expand the therapeutic armamentarium against HPV related anogenital dysplasia are needed.

Artesunate, formulated as a suppository, is approved by the World Health Organization (WHO) as first-line treatment for acute malaria in children who are in remote settings with limited access to healthcare [5]. In addition to the parasiticidal effects in malaria, artemisinin and its derivatives also possess other pharmaceutical properties including antiviral and anti-cancer activity [6–8]. *In vitro* and *in vivo* studies have evaluated artemisinin compounds against infections with enveloped and non-enveloped double-stranded DNA viruses and a variety of solid and hematologic cancers. Artesunate has been shown to be cytotoxic to epithelial cells expressing HPV16 oncogenes E6 and/or E7, while having little effect on uninfected cells [7, 9]. Both anal HSIL and ASCC are associated with functionally obligate expression of these two oncogenes. Epithelial cells expressing E6 and/or E7 overexpress the transferrin receptor, leading to increased intracellular iron levels, compared to normal cells. Artesunate contains an endoperoxide bridge that reacts with intracellular ferrous iron to generate free radicals, leading to cell death [10].

Studies have demonstrated cytotoxic effects of artesunate on HPV-infected cells while having minimal effect on healthy cells [6]. This observation raises the possibility of treating pre-invasive HPV disease (i.e., anal HSIL) with topical artesunate, administered as an intra-anal suppository. The toxicity profile of this formulation is well-documented and includes dizziness, nausea, emesis, and abdominal pain. These symptoms are also commonly experienced in active malarial infection, so symptoms attributed to treatment may be of the disease being treated [11]. Data from a completed study of intravaginal administration of artesunate suppositories for cervical dysplasia has demonstrated that vaginal artesunate inserts to treat CIN 2/3 are safe and well-tolerated [12].

This open label Phase 1 study investigated the safety and tolerability of a novel non-surgical treatment of HPV-associated anal HSIL, using artesunate suppositories.

## Materials and methods

### Patient selection and data collection

A Phase 1 dose escalation trial of intra-anal artesunate in patients with anal HSIL was conducted (NCT03100045). Artesunate suppositories were approved for the treatment of anal HSIL by the Food and Drug Administration through an investigator-initiated Investigational New Drug Application (IND 134720). The Johns Hopkins Hospital Institutional Review Board (IRB00090922) was the single IRB for this study. Patients were recruited from the High-Resolution Anoscopy (HRA) Clinics at The Johns Hopkins Hospital and the University of Wisconsin Hospitals and Clinics. All HRA clinic patients with a history of anal HSIL were offered enrollment into this study. Written consent was obtained from all patients prior to screening for eligibility and enrollment in this clinical trial.

The inclusion criteria were adults who had a positive anal HPV test with anal HSIL biopsies on HRA (Table 1). Patients were enrolled from June 8, 2017 to December 16, 2020, and followed for safety, tolerability and efficacy outcomes with a cutoff date of June 23, 2021 for data analysis (Figs 1 and 2: Study Schema and Timeline). Due to the requirement of administration of artesunate via serial doses and follow-up, the authors had access to information that could identify individual participants during or after data collection.

## Intervention and mode of delivery (Figs 1 and 2)

Participants with anal HSIL biopsies in the previous 8 weeks were screened for study participation. At the time of signing consent, the presence of residual HSIL was documented by HRA. The time between screening for study eligibility and week 0 (study commencement) was ≤ 4–6 weeks.

Artesunate suppositories (Frantz Viral Therapeutics, LLC) were administered intra-anally through a 3+3 dose escalation schema (doses: 200 mg, 400 mg, 600 mg) in which the maximal tolerated dose was identified, with the safeguard of an integrated de-escalation schema. The lowest dose level was two 5-day cycles of artesunate, followed by three five-day cycles of artesunate. In the absence of toxicity, patients were enrolled at the next highest dose in the same manner.

During the week 0 clinic visit, the first suppository of the entire treatment regimen was inserted digitally into the anal canal by the healthcare provider, while the patient was in the left lateral position. Patients were instructed to insert the remaining suppositories at bedtime using the same technique shown by the clinician.

The study team performed a follow-up phone call the week after each treatment cycle began to document compliance with the study medication and any adverse events. At weeks 2, 4, 6, the anoscopic exam assessed for the presence of local mucosal reactions, and symptom diary cards were reviewed, graded, and causality to the study drug assessed by the clinician. Initial screening and follow-up (weeks 16, 28, 40) studies included anal cytology, HPV genotyping (AmpFire HPV genotyping assay, Atila Biosystems, Inc, Sunnyvale, CA), and HRA. HRA-directed biopsies were obtained at week 16, and if clinically indicated, at weeks 28 and 40.

The primary outcome/endpoint was to evaluate the safety and tolerability of intra-anal artesunate. Safety reporting consisted of all adverse events (AEs) reported by patients during the

**Table 1. Inclusion and exclusion criteria.**

| Inclusion Criteria | Exclusion Criteria |
|---|---|
| • Age ≥ 18 years<br>• Biopsy-confirmed high-grade intra-anal dysplasia (AIN 2, AIN 3, HSIL) by HRA including patients who are newly diagnosed with anal HSIL as well as those who have recurrent anal HSIL after medical therapy or surgical therapy<br>• HPV-positive anal test<br>• Use of standard barrier protection use for males and females of reproductive potential for the duration of the study.<br>• Women of childbearing potential: negative urine pregnancy test at screening<br>• Patients who have and have not been immunized with the HPV vaccine<br>• Weight ≥50 kg.<br>• Able to provide informed consent<br>• Patients who have the ability to collaborate with planned follow-up (transportation, compliance history, etc.) | • Diagnosis of low-grade anal dysplasia (AIN 1, LSIL) by HRA, if HSIL was absent<br>• HPV-negative anal test<br>• Known anal, vulvar, cervical, or penile cancer<br>• CD4 count < 200 cells/mm3 at screening for this study. Patients, whose CD4 counts drop < 200 cells/mm3 at any timepoint in this study will be treated with the standard of care treatment arm of surgical ablation<br>• Currently receiving systemic chemotherapy or radiation therapy for another cancer<br>• Patients who are on medical treatment with systemic immunosuppression or steroids (e.g., active autoimmune disease)<br>• Extensive anal condyloma prevents the ability for the clinician to visualize HSIL during HRA<br>• Pregnant female |

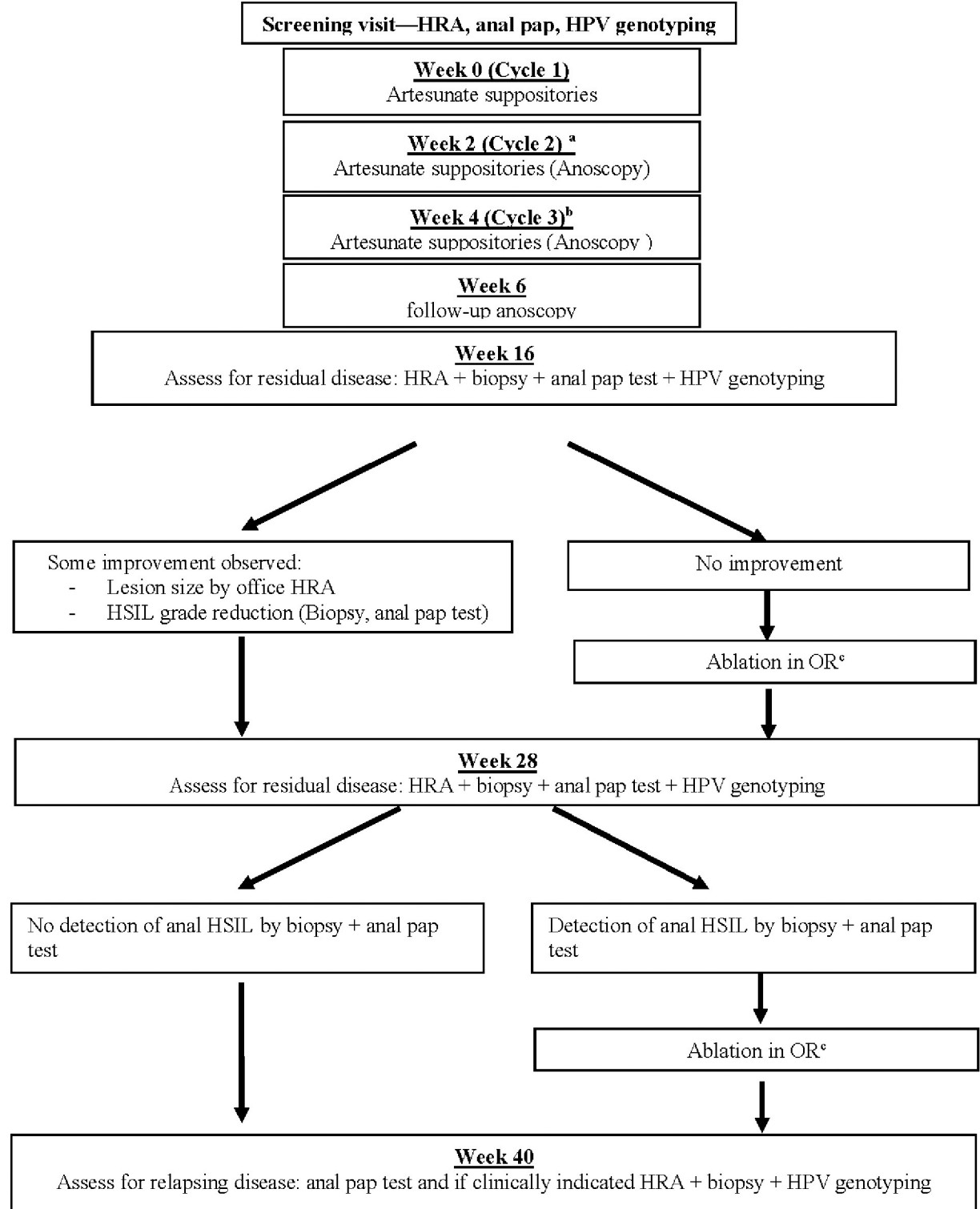

**Fig 1. Study schema and timeline.** CONSORT diagram of a phase 1 study utilizing artesunate for the treatment of anal HSIL. [a]All treatment groups. [b]Only treatment groups 2, 4, and 6 receive suppositories at this visit. [c]According to the standard of care clinical practice, a postoperative visit will be done 4 weeks after ablation, i.e. either on week 20 or on week 32.

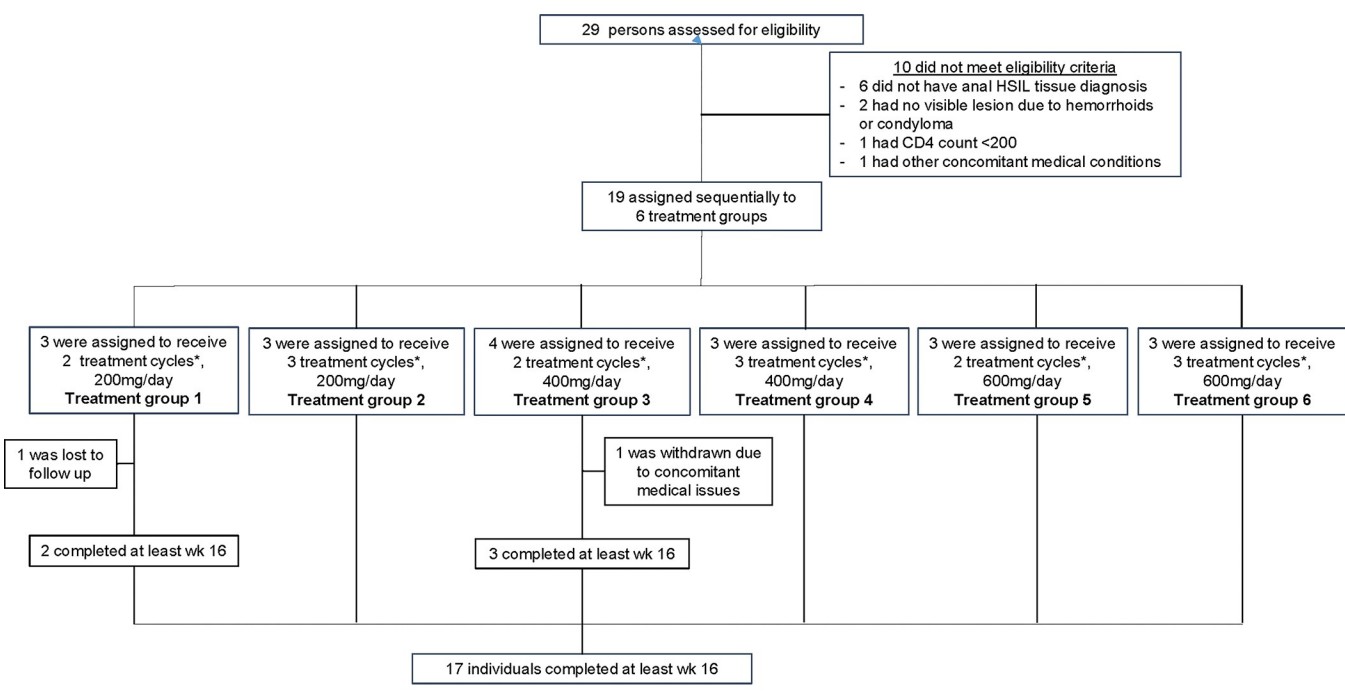

**Fig 2. Enrollment schema.** * each treatment cycle consisted of five (5) consecutive nightly artesunate suppositories.

dosing phase of the study and for one month following the last dose of Artesunate. Relatedness of an AE to study treatment was assessed by the investigator.

A dose-limiting toxicity was defined as any drug-related Grade 2 or greater, drug-related toxicities in any organ system as delineated in Common Terminology Criteria for Adverse Events v4.0. Tolerability was defined as the percentage of participants who completed the dosing regimen without any drug-related serious adverse event (SAE).

Secondary outcomes/endpoints for efficacy included the following:

1. Complete histologic regression is defined as either regression to LSIL (low-grade squamous intraepithelial lesion) or no anal dysplasia detected by HRA/biopsy and anal cytology at week 16 and over the study window. Partial regression is defined as: (1) clearance of intra-anal HSIL, but not perianal HSIL in a patient who had both or (2) >50% reduction of the area of HSIL lesions compared to the area of anal HSIL seen on screening HRA

2. Clearance of detectable HPV genotypes identified at the screening evaluation that became undetectable consistently through serial HPV genotyping over the study window. New genotypes acquired during the study were not included in evaluation.

Descriptive summary statistics were used to assess demographic characteristics, safety, and regression of anal HSIL. Categorical data were summarized using frequencies and relative frequencies (i.e., proportions). Continuous data were summarized by mean or median with the range.

## Results

### Baseline patient characteristics (Fig 1 and Table 2)

- **Screened population**: all patients who signed informed consent, regardless of enrollment status (29 patients).

**Table 2. Baseline characteristics of enrolled study populations.**

| | HIV+<br>N = 7 | HIV-<br>N = 12 |
|---|---|---|
| Age (median, range) | 44.9 (range 26–69) | 48.2 (range 28–69) |
| Gender, N (%) male | 6/7 (85.7%) | 6/12 (50%) |
| MSM status (N, %) | 6/7 (85.7%) | 6/12 (50%) |
| CD4 count (cell count, median range) | 655 cells/mm$^3$<br>(range 467–1302) | n/a |
| ART therapy, N (%) | 7 (100%) | n/a |
| ART therapy duration | 17 years (range 4–30) | n/a |
| Smoking | | |
| previous | 4 (57%) | 6 (50%) |
| current | 3 (43%) | 1 (8%) |
| Number of sexual partners (median, range) | 32 (range 5–100) | 39 (range 3–200) |
| Prevalent HPV types at screening | HPV16 (86%)<br>HPV18 (29%)<br>other high-risk HPV (100%) | HPV16 (67%)<br>other high-risk HPV 7 (58%) |
| | 1 HPV subtype (17%)<br>>1 HPV subtype (83%) | 1 HPV subtype (75%)<br>>1 HPV subtype (25%) |
| Previous anal HSIL treatment (number of patients, %) | 4 (57%) | 6 (50%) |
| 40-week follow-up visit completed | 6/7 (85.7%) | 9/12 (75%) |

The following patient groups describe the populations analyzed in this study

- **Safety analysis population**–all participants who received at least one dose of artesunate, regardless of study completion (19 patients).

- **Modified intention to treat population** (mITT)–participants who received at least one dose of artesunate, for whom efficacy endpoint data are available (17 patients).

In total, 29 patients were screened for eligibility. Ten failed screening due to having no anal HSIL on HRA biopsy or a negative HPV test (Fig 1). A total of nineteen participants with anal HSIL (median age 49 years; range, 26–69 years) were assigned to 6 treatment groups and received intra-anal artesunate. All 12 (63%) males enrolled were men who have sex with men (MSM). Fifteen participants (79%) reported a history of anoreceptive sex. Seven participants were PLWH (37%) (1 female, 6 males). The average CD4 count was 655 cells/mm$^3$ (range 467–1302). Most had an undetectable viral load (6/7, 86%). One had a viral load of 8430 copies/ml. HPV16 was the most prevalent type for both PLWH (6/7, 86%) and HIV-negative patients (7/12; 58%). The majority of PLWH had > 1 HPV genotype (6/7, 86%), whereas a smaller proportion of HIV-negative individuals had > 1 HPV genotype (5/12, 42%). Two patients were included in the safety analysis, but not in the mITT population. In Group 1, one patient was non-compliant with follow-up and in Group 3, one patient was removed from the study due to non-related medical health issues.

## Safety and tolerability

Eighteen out of nineteen patients enrolled in this study completed the full dosing regimens and artesunate treatment was generally well tolerated. One patient did not complete the full dosing regimen due to a mental health issue. A total of 119 AEs were reported in 15 out of the 19 patients who were treated with artesunate [Table 3: Adverse Events]. Fifty-nine AEs were deemed to be potentially drug-related by the investigator.

**Table 3. Adverse events.**

| Cohort | Category | | Grade | Occurrences |
|---|---|---|---|---|
| **200mg, 2 cycles** | Gastrointestinal disorders | Abdominal pain | I | 1 |
| | | Anal pain | I | 3 |
| | | Nausea | I | 2 |
| | | Diarrhea | I | 1 |
| | Nervous system disorders | Dizziness | I | 2 |
| | | Headache | I | 3 |
| **200mg, 3 cycles** | Gastrointestinal disorders | Abdominal pain | I | 2 |
| | | Diarrhea | I | 2 |
| | | Flatulence | I | 1 |
| | | Nausea | I | 1 |
| | Nervous system disorders | Paresthesia | I | 1 |
| | Skin and subcutaneous tissue disorders | Pain of skin | I | 5 |
| **400mg, 2 cycles** | Gastrointestinal disorders | Anal spasm | I | 1 |
| | | Nausea/vomiting | I | 2 |
| | Nervous system disorders | Headache | I | 2 |
| **400mg, 3 cycles** | Gastrointestinal disorders | Anal irritation | I | 2 |
| | | Anal pain | I | 3 |
| | | Constipation | I | 1 |
| | | Diarrhea | I | 6 |
| | | Dry mouth | I | 1 |
| | | Flatulence | I | 1 |
| | | Nausea/vomiting | I | 2 |
| | General disorders and administration site conditions | Fatigue | I | 2 |
| | Injury, poisoning and procedural complications | Bruising | I | 1 |
| | Musculoskeletal and connective tissue disorders | Low back pain | I | 1 |
| | | Low back pain | II | 1 |
| | | Myalgia | I | 1 |
| | Nervous system disorders | Dizziness | I | 2 |
| | | Headache | I | 2 |
| | | Headache | II | 1 |
| | Skin and subcutaneous tissue disorders | Anal Pruritus | I | 3 |
| **600mg, 2 cycles** | Ear and labyrinth disorders | Vertigo | I | 1 |
| | Gastrointestinal disorders | Abdominal pain | I | 1 |
| | | Anal hemorrhage | I | 3 |
| | | Anal mucositis | I | 1 |
| | | Anal pain | I | 6 |
| | | Flatulence | I | 1 |
| | | Nausea | I | 8 |
| | General disorders and administration site conditions | Fatigue | I | 2 |
| | | Fever | I | 1 |
| | Infections and infestations | Esophageal infection | II | 1 |
| | | Toenail fungus | I | 1 |
| | Nervous system disorders | Headache | I | 1 |
| | Psychiatric disorders | Anxiety | I | 1 |
| | Respiratory, thoracic, and mediastinal disorders | Sore throat | I | 1 |
| | Skin and subcutaneous tissue disorders | Rash, maculo papular | I | 1 |

*(Continued)*

**Table 3.** (Continued)

| Cohort | Category | | Grade | Occurrences |
|---|---|---|---|---|
| **600mg, 3 cycles** | Gastrointestinal disorders | Abdominal pain | I | 1 |
| | | Anal hemorrhage | I | 5 |
| | | Anal itching | I | 4 |
| | | Anal pain | I | 1 |
| | | Flatulence | I | 1 |
| | | Nausea/vomiting | I | 3 |
| | | Nausea/vomiting | II | 4 |
| | General disorders and administration site conditions | Fatigue | I | 4 |
| | Metabolism and nutrition disorders | Diabetic ketoacidosis | IV | 1 |
| | Nervous system disorders | Dizziness | I | 4 |
| | | Headache | I | 1 |
| | Skin and subcutaneous tissue disorders | Perianal pruritus | I | 1 |

All AEs associated with the study drug were Grades 1 or 2. Patients in treatment groups receiving 200- and 400-mg suppositories reported a limited number of grade 1 AE's related to the study medication which were not dose limiting. However, most participants enrolled in the 600 mg arm, reported Grade 1 AEs of nausea and one reported grade 2 nausea, which was controlled with ondansetron. Nausea was considered dose-limiting at the 600 mg dose. No SAEs associated with the study drug were reported by participants in any treatment group. One patient had a non-related SAE (diabetic ketoacidosis) due to non-compliance with insulin treatment.

## Efficacy

**Histologic regression ([Table 4]).** At screening HRA, all study participants had residual anal HSIL as documented by HRA, which ranged from minimal disease (1 or 2 small lesions, < 5 mm in size) to extensive intra-anal and perianal disease.

A total of 10/17 (59%) HSIL experienced either complete (6/17, 35%) or partial (4/17, 24%) regression identified at the week 16, 28, and week 40 follow-up HRAs. For those patients who continued to have anal HSIL on their week 16 biopsy, they were offered surgery and all chose to postpone their surgery until later in the clinical trial.

**Clearance of HPV.** Three (50%) of the 6 patients who experienced complete histologic regression had clearance of the HPV types that were detected at baseline. Of note, all of them carried non-HPV16 high-risk types. One participant who had HPV clearance was living with HIV.

## Subgroup analyses

One out of six (17%) PLWH in the mITT population had complete regression of anal HSIL; whereas, 8/11 (73%) HIV-negative patients had either partial [3/11 (27%)] or complete [5/11 (46%] regression of their anal HSIL.

The highest regression rate occurred in participants who had non-HPV16 types (4/5, 80%), followed by participants who had HPV16 mono-infection (3/4, 75%). Regression rates were lower in those who had HPV16 and other high-risk types (2/8, 25%).

## Discussion

In May 2020, the FDA approved intravenous artesunate as first-line treatment for severe malaria, in infants, children, and adults. Between 2000–2020, 10.6 million malarial deaths were

**Table 4. Treatment group and regression.**

| ID | Gender[a] | Race[b] | Age | HIV status | HPV 16? | Other HPV subtypes | Extent of HSIL | Response[c] | HR (wks) | VC | VC (wks) |
|---|---|---|---|---|---|---|---|---|---|---|---|
| ART200_2_1 | F | W | 36 | Negative | Y | | >50% | Recurrence | | | |
| ART200_2_2 | F | B | 30 | Positive | Y | 11,31,40,42,45,58,73,84, CP6108 | >50% | PR-removed from study | | | |
| ART200_2_3 | M | B | 36 | Positive | Y | 33,45,51,52,53,56,58,66,70 | >50% | PR | | | |
| ART200_3_1 | M | W | 47 | Negative | N | 53,61,69,73,CP6108 | >50% | CR | 16 | Y | 40 |
| ART200_3_2 | M | W | 28 | Negative | Y | 42 | <50% | PR | | | |
| ART200_3_3 | F | W | 69 | Negative | Y | | <50% | CR | 16 | | |
| ART400_2_1 | M | B | 59 | Positive | Y | 52,56,66 | >50% | NR | | | |
| ART400_2_2 | F | W | 32 | Negative | Y | | >50% | PR | | | |
| ART400_2_3 | M | W | 58 | Negative | Y | | <50% | CR | 16 | | |
| ART400_3_1 | M | B | 52 | Positive | Y | 53,58,62,66,84 | <50% | NR | | | |
| ART400_3_3 | M | W | 57 | Negative | N | 45 | <50% | CR | 16 | Y | 6 |
| ART400_3_4 | M | W | 61 | Positive | Y | 18,59 | >50% | NR | | | |
| ART600_2_1 | M | B | 42 | Positive | N | 68 | <50% | CR | 16 | Y | 40 |
| ART600_2_2 | M | W | 26 | Positive | Y | 18,33,53,58,66 | <50% | NR | | | |
| ART600_2_3 | F | W | 58 | Negative | Y | 31 | >50% | NR | | | |
| ART600_3_1 | F | W | 60 | Negative | Y | | >50% | PR | | | |
| ART600_3_2 | M | Multi | 32 | Negative | N | 31,51,53 | <50% | NR | | | |
| ART600_3_3 | M | W | 40 | Negative | Y | 31,39 | <50% | CR | 16 | | |

[a]Gender

M: Male

F: Female

[b]Race

W: White

B: Black

Multi: multi-racial

[c]Response

PR: partial regression

CR: complete regression

NR: No regression

VC: Viral (HPV) clearance

averted due to the utilization of artesunate combination therapy [5]. The safety, tolerability, and pharmacokinetics are well-characterized, based on clinical experience in over 7 million acutely ill persons [11]. This study confirms the safety and tolerability of artesunate suppositories in the treatment of anal HSIL.

Over the past years, office-based ablation via electrocautery has been used effectively in many HRA clinics and was the main treatment modality in the ANCHOR study [2]). However, an experienced HRA treatment center recently described an overall HSIL recurrence rate of 50% within 1 year and 68% within 3 years of ablation, despite initial response in 62% of patients after single ablation treatment [3]. Microscopic residual disease and multiple index HSIL lesions as well as persistent HPV 16/18 infection may increase risk of recurrence [3].

Topical treatments of anal HSIL are preferable for some patients due to ease of use and less invasiveness. Off-label topical treatments of anal HSIL include 5-FU and more commonly the TLR-7 agonists imiquimod, which stimulates anti-viral cytokines and natural killer cell responses against HPV-infected cells [13]. Treatment response to imiquimod and 5-FU were lower in an open-label randomized controlled trial in MSM living with HIV as compared to

electrocautery (overall complete and partial response rates for imiquimod, 5-FU, and electro-cautery were 46%, 42%, and 69%, respectively) and Grade 3–4 adverse events common in the topical treatment groups (43% in the imiquimod and 27% in the 5-FU group), [14, 15]. How-ever, a recent small, single center, longitudinal cohort study of MSM living with HIV and diag-nosed with HSIL found excellent response rates and tolerability of imiquimod with less recurrence compared to surgical treatment [4]. In a systematic review of 14 studies of intra-anal imiquimod, complete response in 211 patients with HSIL was observed in 35%, partial response 20.9% and recurrence in 15% [16]. Differences in the response rates may be related to patient factors such as extent of disease, degree of immunosuppression or other risk factors such as smoking but also to the dose of imiquimod or the formulation (cream vs suppository versus tampon) [16].

In the current study, 59% of participants treated with artesunate suppositories had a partial or complete response over the study window, and 50% of the complete responders also had viral clearance. While the primary objective of this Phase 1 study was to evaluate safety and tol-erability, data suggest that artesunate could be effective in the treatment of intra-anal HSIL. These findings are similar to the phase I study of artesunate vaginal inserts for the treatment of CIN2/3, where 67.9% had complete response, and 47.4% of those with complete response also underwent viral clearance [12]. In this study, histologic regression preceded HPV clearance in two thirds of patients who experienced both regression and viral clearance, In combination with dose response observations, the authors concluded that artesunate cytotoxicity is medi-ated by direct contact; cell death and cytolysis may subsequently induce a pro-inflammatory milieu and adaptive anti-HPV T cell responses as cell content becomes visible to the immune system. Whether the mode of action of artesunate is similar in anal disease remains to be evaluated.

With the limitation of the small cohort size, subgroup analyses suggest that artesunate may be more effective in the treatment of anal HSIL of HIV negative patients versus PLWH, who are known to have a higher risk of HSIL recurrence and HSIL to ASCC progression [15]. While the small sample size and limited statistical power precludes definitive interpretation of therapeutic response data, this observation would be consistent with experience of other treat-ment modalities for anal HSIL described above.

Once a MTD was determined to be 400mg, the study sponsor decided to further evaluate key secondary outcomes in a controlled Phase IIB study. A randomized, placebo-controlled phase IIB study of artesunate suppositories for HIV-negative individuals with intra-anal HSIL is currently open to enrollment. A second controlled Phase IIB study in HIV+ individuals will open soon. Evaluation of the treatment effect of intra-anal artesunate will be the primary focus of the ongoing Phase 2 trial in anal HSIL. Furthermore, clinical trials are ongoing to evaluate the efficacy of topical artesunate in cervical and vulvar HPV-related disease (clinicaltrials.gov: NCT03792516, NCT04098744, NCT05555862)

A potential concern with topical artesunate therapy for treatment of HPV-related disease is that it might contribute to malarial resistance if used for patients in malaria endemic regions of the world. Resistance of *Plasmodium falciparum* arises from several factors, including improper use or overuse of antimalarial drugs for prophylaxis, inadequate or incomplete ther-apeutic treatments of active infections, a high level of parasite adaptability at the genetic and metabolic levels, and a high parasite proliferation rate that permits selected populations to emerge rapidly [17]. It is unlikely that a patient with symptomatic acute malaria would be started on intra-anal artesunate for anal dysplasia [16]; however, asymptomatic parasitemia may have to be addressed before treatment.

The use of intra-anal artesunate to treat anal HSIL is not anticipated to contribute to global malarial resistance development, as patients acutely sick with acute malaria would be unlikely

to receive simultaneous treatment for anal disease. Systemic absorption of the topical artesunate formulation used in this study will be evaluated during Phase 2; this information will provide further information to assess this risk, including in the setting of asymptomatic parasitemia in malaria endemic regions.

## Conclusions

Intra-anal artesunate is a safe and well-tolerated treatment for anal HSIL and may be a promising addition in the therapeutic armamentarium against HPV-related anogenital disease. It is easily self-administered in the suppository form. Due to the frequent nausea that patients experienced at the 600-mg dose, the placebo-controlled, randomized Phase 2B trial will utilize the 400-mg suppository. We would like to thank Dr. Namandje Bumpus and Dr. Craig Hendrix in the Division of Clinical Pharmacology with the Department of Internal Medicine for their contributions to the study design.

## Supporting information

**S1 Checklist. Reporting checklist for randomised trial.**
(DOCX)

**S2 Checklist. TREND statement checklist.**
(PDF)

**S1 Dataset.**
(XLSX)

**S1 Protocol.**
(DOC)

## Acknowledgments

We would like to thank Dr. Namandje Bumpus and Dr. Craig Hendrix in the Division of Clinical Pharmacology with the Department of Internal Medicine for their contributions to the study design.

## Author Contributions

**Conceptualization:** Sandy Hwang Fang, Mihaela Plesa, Emily Staudt, Cornelia L. Trimble.

**Data curation:** Sandy Hwang Fang, Mihaela Plesa, Evie H. Carchman, Nicole A. Cowell, Emily Staudt, Kyleigh Ann Twaroski, Ulrike K. Buchwald, Cornelia L. Trimble.

**Formal analysis:** Sandy Hwang Fang, Mihaela Plesa, Nicole A. Cowell, Emily Staudt, Ulrike K. Buchwald, Cornelia L. Trimble.

**Funding acquisition:** Sandy Hwang Fang, Mihaela Plesa, Cornelia L. Trimble.

**Investigation:** Sandy Hwang Fang, Mihaela Plesa, Evie H. Carchman, Nicole A. Cowell, Emily Staudt, Kyleigh Ann Twaroski, Ulrike K. Buchwald, Cornelia L. Trimble.

**Methodology:** Sandy Hwang Fang, Mihaela Plesa, Nicole A. Cowell, Emily Staudt, Ulrike K. Buchwald, Cornelia L. Trimble.

**Project administration:** Sandy Hwang Fang, Mihaela Plesa, Evie H. Carchman, Nicole A. Cowell, Emily Staudt, Kyleigh Ann Twaroski, Ulrike K. Buchwald.

**Resources:** Sandy Hwang Fang, Mihaela Plesa, Evie H. Carchman, Nicole A. Cowell, Emily Staudt, Kyleigh Ann Twaroski, Ulrike K. Buchwald.

**Software:** Sandy Hwang Fang.

**Supervision:** Sandy Hwang Fang, Mihaela Plesa, Evie H. Carchman, Emily Staudt, Kyleigh Ann Twaroski.

**Validation:** Sandy Hwang Fang, Mihaela Plesa, Evie H. Carchman, Nicole A. Cowell, Emily Staudt.

**Visualization:** Sandy Hwang Fang.

**Writing – original draft:** Sandy Hwang Fang, Mihaela Plesa, Nicole A. Cowell.

**Writing – review & editing:** Sandy Hwang Fang, Mihaela Plesa, Evie H. Carchman, Nicole A. Cowell, Emily Staudt, Kyleigh Ann Twaroski, Ulrike K. Buchwald, Cornelia L. Trimble.

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
