## [Decision Letter · Decision Letter 0]

13 Jun 2023

PONE-D-23-09503A Phase I Study of Intra-anal Artesunate (suppositories) to Treat Anal High-grade Squamous Intraepithelial LesionsPLOS ONE

Dear Dr. Fang,

Thank you for submitting your manuscript to PLOS ONE. After careful consideration, we feel that it has merit but does not fully meet PLOS ONE’s publication criteria as it currently stands. Therefore, we invite you to submit a revised version of the manuscript that addresses the points raised during the review process.

We look forward to receiving your revised manuscript.

Kind regards,

Guglielmo Campus, Ph.D DDS

Academic Editor

PLOS ONE

Comments from PLOS Editorial Office: We note that one or more reviewers has recommended that you cite specific previously published works. As always, we recommend that you please review and evaluate the requested works to determine whether they are relevant and should be cited. It is not a requirement to cite these works. We appreciate your attention to this request.

“The competing interest exists in the fact that Mihaela Plesa became the Director of Research at Frantz Viral Therapeutics when she left Johns Hopkins Hospital.  Please note that we had already written the protocol, obtained FDA and IRB approval, and started the study prior to Ms. Plesa leaving Hopkins to join Frantz Viral Therapeutics.”

5. Please include your tables as part of your main manuscript and remove the individual files. Please note that supplementary tables (should remain/ be uploaded) as separate "supporting information" files.

7. We note that the original protocol file you uploaded contains a confidentiality notice indicating that the protocol may not be shared publicly or be published. Please note, however, that the PLOS Editorial Policy requires that the original protocol be published alongside your manuscript in the event of acceptance. Please note that should your paper be accepted, all content including the protocol will be published under the Creative Commons Attribution (CC BY) 4.0 license, which means that it will be freely available online, and any third party is permitted to access, download, copy, distribute, and use these materials in any way, even commercially, with proper attribution.

Therefore, we ask that you please seek permission from the study sponsor or body imposing the restriction on sharing this document to publish this protocol under CC BY 4.0 if your work is accepted. We kindly ask that you upload a formal statement signed by an institutional representative clarifying whether you will be able to comply with this policy. Additionally, please upload a clean copy of the protocol with the confidentiality notice (and any copyrighted institutional logos or signatures) removed.

8. We note that the original protocol that you have uploaded as a Supporting Information file contains an institutional logo. As this logo is likely copyrighted, we ask that you please remove it from this file and upload an updated version upon resubmission.

Reviewers' comments:

Reviewer's Responses to Questions

**Comments to the Author**

1. Is the manuscript technically sound, and do the data support the conclusions?

Reviewer #1: Yes

Reviewer #2: Yes

2. Has the statistical analysis been performed appropriately and rigorously? 

Reviewer #1: Yes

Reviewer #2: Yes

3. Have the authors made all data underlying the findings in their manuscript fully available?

Reviewer #1: Yes

Reviewer #2: Yes

4. Is the manuscript presented in an intelligible fashion and written in standard English?

Reviewer #1: Yes

Reviewer #2: Yes

5. Review Comments to the Author

Reviewer #1: The authors conducted a 3+3 dose escalation Phase I clinical trial to evaluate the safety and tolerability of artesunate suppositories in the treatment of patients with biopsy-proven HSIL. Nineteen patients were enrolled for evaluation, and the maximal tolerated dose was determined at 400 mg, administered in 3 cycles. Overall, the manuscript was well written. I have a few comments on the statistical analysis and evaluation.

1. After determine the maximal tolerated dose at 400mg, there is no extension cohort at this dose level to evaluate the key secondary outcomes such as complete histologic regression, clearance of dateable HPV genotypes. Need to put into the discussion for the limitation.

2. For the participants enrolled in the 600 mg arm, only one experienced grade 2 nausea, then the authors decided that Nausea was considered dose-limiting at the 600 mg dose, why not enroll 3 more participants into the 600 mg arm following the standard 3+3 scheme?

Reviewer #2: The study is an early phase trial on the efficacy of artesunate suppository for anal HSIL.El estudio se trata de un ensayo en fase precoz sobre la eficacia del artesunato en supositorio para HSIL anal. es interesante y novedoso, y presenta otra opción en el tratamiento de la displasia anal de alto grado. One of the points that concerns me in this study is that more than 50% of the bibliographic references, to be exact 56.25% are more than 10 years old, some of them close to 20 years old. I recomend to read and includ this reseacrh for example: Hidalgo-Tenorio C,et al. Risk factors for ≥high-grade anal intraepithelial lesions in MSM living with HIV and the response to topical and surgical treatments One. 2021 Feb 3;16(2):e0245870; or this one: Fuertes I, et al The effectiveness and tolerability of imiquimod suppositories to treat extensive intra-anal high-grade squamous intraepithelial lesions/warts in HIV-infected individuals. Int J STD AIDS. 2019 Oct;30(12):1194-1200. The discussion is very poorly elaborated, it should be ostensibly improved if it is finally published.

6. PLOS authors have the option to publish the peer review history of their article (what does this mean?). If published, this will include your full peer review and any attached files.

Reviewer #1: No

Reviewer #2: No

---

## [Author Response · Author response to Decision Letter 0]

31 Jul 2023

PONE-D-23-09503

A Phase I Study of Intra-anal Artesunate (suppositories) to Treat Anal High-grade Squamous Intraepithelial Lesions

PLOS ONE

Dear Reviewers, 

Thank you for your thorough review of our manuscript.

Author Response

The manuscript has been reformatted to meet the PLOS ONE’s style requirements.

Author Response

The grant information in the “funding information” and the “financial disclosure” now match.

For grant numbers, we currently do not have a grant number for the compensation we received for our study, as this is an industry-sponsored study.

“The competing interest exists in the fact that Mihaela Plesa became the Director of Research at Frantz Viral Therapeutics when she left Johns Hopkins Hospital. Please note that we had already written the protocol, obtained FDA and IRB approval, and started the study prior to Ms. Plesa leaving Hopkins to join Frantz Viral Therapeutics.”

Author Response

We have updated our competing interests statement in our cover letter and have also included this in our manuscript under financial disclosure and competing interests. Thank you.

Author response:

We have included the figure caption in the manuscript.

5. Please include your tables as part of your main manuscript and remove the individual files. Please note that supplementary tables (should remain/ be uploaded) as separate "supporting information" files.

Author Response:

We have included all tables in the main manuscript document, and removed them as separate files. 

Author Response:

We have included our minimal data set in a supporting information file

7. We note that the original protocol file you uploaded contains a confidentiality notice indicating that the protocol may not be shared publicly or be published. Please note, however, that the PLOS Editorial Policy requires that the original protocol be published alongside your manuscript in the event of acceptance. Please note that should your paper be accepted, all content including the protocol will be published under the Creative Commons Attribution (CC BY) 4.0 license, which means that it will be freely available online, and any third party is permitted to access, download, copy, distribute, and use these materials in any way, even commercially, with proper attribution.

Therefore, we ask that you please seek permission from the study sponsor or body imposing the restriction on sharing this document to publish this protocol under CC BY 4.0 if your work is accepted. We kindly ask that you upload a formal statement signed by an institutional representative clarifying whether you will be able to comply with this policy. Additionally, please upload a clean copy of the protocol with the confidentiality notice (and any copyrighted institutional logos or signatures) removed.

Author Response:

We have uploaded the protocol without the confidentiality notice and without the institutional logo. Please note that this protocol was updated and approved by Mihaela Plesa, who is the acting official and Director of Research for Frantz Viral Therapeutics. 

8. We note that the original protocol that you have uploaded as a Supporting Information file contains an institutional logo. As this logo is likely copyrighted, we ask that you please remove it from this file and upload an updated version upon resubmission.

Author Response:

We have uploaded the protocol without the institutional logo. 

Comments to the Author

Reviewer #1: The authors conducted a 3+3 dose escalation Phase I clinical trial to evaluate the safety and tolerability of artesunate suppositories in the treatment of patients with biopsy-proven HSIL. Nineteen patients were enrolled for evaluation, and the maximal tolerated dose was determined at 400 mg, administered in 3 cycles. Overall, the manuscript was well written. I have a few comments on the statistical analysis and evaluation.

1. After determine the maximal tolerated dose at 400mg, there is no extension cohort at this dose level to evaluate the key secondary outcomes such as complete histologic regression, clearance of dateable HPV genotypes. Need to put into the discussion for the limitation.

Author response:

Once a MTD was determined to be 400mg, the study sponsor decided to further evaluate key secondary outcomes in a controlled Phase IIB study. A randomized, placebo-controlled phase IIB study of artesunate suppositories for HIV-negative individuals with intra-anal HSIL is currently open to enrollment. A second controlled Phase IIB study in HIV+ individuals will open soon.

This has been added to the discussion section of the paper.

2. For the participants enrolled in the 600 mg arm, only one experienced grade 2 nausea, then the authors decided that Nausea was considered dose-limiting at the 600 mg dose, why not enroll 3 more participants into the 600 mg arm following the standard 3+3 scheme?

Author response:

The reason why we decided not to proceed with the 600-mg dose was because we felt that despite the nausea being graded as a 2, we felt that the nausea was too intense for the intent of treatment. We did not think that it would be appropriate for our patients to be subjected to this side effect. We decided upon the highest dose level with the least number of AEs. We were hoping that given that our patient did not experience nausea in the 400-mg group that we would further evaluate efficacy of the 400-mg dose in the phase IIB study. 

Reviewer #2: The study is an early phase trial on the efficacy of artesunate suppository for anal HSIL. El estudio se trata de un ensayo en fase precoz sobre la eficacia del artesunato en supositorio para HSIL anal. es interesante y novedoso, y presenta otra opción en el tratamiento de la displasia anal de alto grado. One of the points that concerns me in this study is that more than 50% of the bibliographic references, to be exact 56.25% are more than 10 years old, some of them close to 20 years old. I recomend to read and includ this reseacrh for example: Hidalgo-Tenorio C,et al. Risk factors for ≥high-grade anal intraepithelial lesions in MSM living with HIV and the response to topical and surgical treatments One. 2021 Feb 3;16(2):e0245870; or this one: Fuertes I, et al The effectiveness and tolerability of imiquimod suppositories to treat extensive intra-anal high-grade squamous intraepithelial lesions/warts in HIV-infected individuals. Int J STD AIDS. 2019 Oct;30(12):1194-1200. The discussion is very poorly elaborated, it should be ostensibly improved if it is finally published.

Author Response:

Thank you for your recommendation. We have revised the discussion and references to address your comments and include more recent publications, including the Hidalgo-Tenorio paper mentioned above.

---

## [Editor Report · Decision Letter 1]

28 Nov 2023

A Phase I Study of Intra-anal Artesunate (suppositories) to Treat Anal High-grade Squamous Intraepithelial Lesions

PONE-D-23-09503R1

Dear Dr. Fang,

We’re pleased to inform you that your manuscript has been judged scientifically suitable for publication and will be formally accepted for publication once it meets all outstanding technical requirements.

Kind regards,

Diane M Harper, MD, MPH, MS

Academic Editor

PLOS ONE
---

## [Editor Report · Acceptance letter]

7 Dec 2023

PONE-D-23-09503R1 

A phase I study of intra-anal artesunate (suppositories) to treat anal high-grade squamous intraepithelial lesions 

Dear Dr. Fang:

I'm pleased to inform you that your manuscript has been deemed suitable for publication in PLOS ONE. Congratulations! Your manuscript is now with our production department. 

Kind regards, 

on behalf of

Dr. Diane M Harper 

Academic Editor

PLOS ONE